# Non-Cross Diffusion for Semantic Consistency

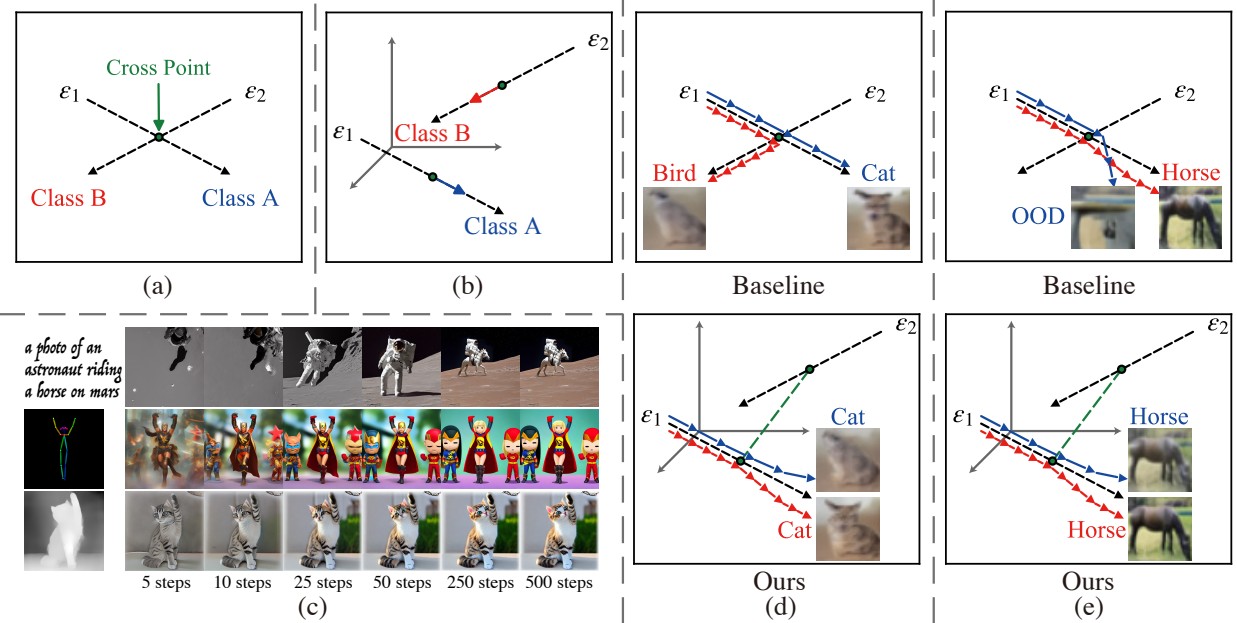

Figure 1: Illustrating xFLOW in Diffusion Models. (a) Demonstrates the ambiguity in training targets caused by crossing flows, leading to the xFLOW problem. (b) Shows how our method eliminates flow crossing by increasing the dimensionality of network inputs, thus resolving the xFLOW problem. (c) Depicts how xFLOW leads to variable sampling results across different steps, undermining deterministic sampling even for Stable Diffusion (Rombach et al., 2022). (d) *Top*: Highlights the discrepancies between outcomes from reduced steps sampling (blue) versus standard results (from 1000 steps in red) due to xFLOW. *Bottom*: Our method ensures consistent outputs across different sampling steps. (e) *Top*: Exhibits instances where xFLOW causes Out-Of-Distribution (OOD) outcomes in reduced steps sampling (blue) compared to standard results (from 1000 steps in red). *Bottom*: Our approach minimizes the occurrence of OOD samples.

## Abstract

In diffusion models, deviations from a straight generative flow are a common issue, resulting in semantic inconsistencies and suboptimal generations. To address this challenge, we introduce *Non-Cross Diffusion*, an innovative approach in generative modeling for learning ordinary differential equation (ODE) models. Our methodology strategically incorporates an ascending dimension of input to effectively connect points sampled from two distributions with uncrossed paths. This design ensures enhanced semantic consistency throughout the inference process, which is especially critical for applications reliant on consistent generative flows, including distillation methods and deterministic sampling, which are fundamental in image editing and interpolation tasks. Our empirical results demonstrate the effectiveness of Non-Cross Diffusion, showing a substantial reduction in semantic inconsistencies at different inference steps and a notable enhancement in the overall performance of diffusion models.

# 1 Introduction

Diffusion models, as delineated in recent studies (Song et al., 2020a; Ho et al., 2020; Dhariwal & Nichol, 2021; Nichol & Dhariwal, 2021; Rombach et al., 2022; Song & Ermon, 2019; Song et al., 2020b), have exhibited remarkable capabilities in image synthesis, bolstering numerous applications such as text-to-image generation (Nichol et al., 2022; Saharia et al., 2022), image editing (Avrahami et al., 2022; Nichol et al., 2022; Brooks et al., 2023; Tumanyan et al., 2023), and image inpainting (Avrahami et al., 2022; Ramesh et al., 2022). A key characteristic of these models is their multi-step generative process, which not only allows for correction of the diffusion path (Song et al., 2020b) but also enhances controllability (Fan & Lee, 2023; Gao et al., 2023).

Despite these advancements, the inference process in diffusion models typically involves a specific flow. However, the training process is more complex and involves random step sampling from multiple flows. Such randomness often results in a particular training step being correlated with numerous diverse flows. Specifically, a single training step can be associated with multiple flows. This correlation with diverse flows introduces ambiguity and uncertainty in the target from the optimization's perspective, as depicted in Figure 1(a). We term this phenomenon as 'xFLOW'.

xFLOW's emergence during training can hinder the model's optimization at certain steps, leading to a spectrum of generative issues. Notably, it challenges the model's ability to generate samples via a straight flow, compromising deterministic sampling across varying step counts, as shown in Fig. 1(c). It also complicates predicting later sampling steps from earlier ones, limiting the effectiveness of reward models (Yoon et al., 2023) and guided models (Dhariwal & Nichol, 2021). Moreover, in the context of distillation, which typically adopts a progressive approach, xFLOW can introduce misleading signals, as evidenced in Rectified Flow (Liu et al., 2022). Perhaps most critically, xFLOW can lead to the generation of Out-Of-Distribution (OOD) samples or low-quality samples, especially as sampling step size increases, as illustrated in Fig. 1(d-e).

In this paper, we propose a novel training strategy aimed at resolving the xFLOW challenge in diffusion models. Our method centers on augmenting the input dimensionality to these models, a change that effectively prevents flow crossing. As depicted in Fig. 1(a), the issue at hand arises when two flows intersect, creating ambiguity; the input to the network (for instance, a noisy image) remains constant, yet it is associated with multiple potential targets (such as distinct noises originating from different images). To address this, our approach entails predicting the flow itself during the training phase, as shown in Fig. 1(b). Notably, we utilize the noise predicted by the network as the flow's endpoint, incorporating this element into the model's input. This technique sidesteps the pitfall of using groundtruth noise as input, which would otherwise result in trivial training solutions devoid of substantive learning. For practical implementation, we found ControlNet (Zhang et al., 2023) particularly effective in this context. Additionally, our methodology integrates a bootstrap approach reminiscent of Analog bits (Chen et al., 2022), which significantly enhances our model's optimization and effectively narrows the gap between training and inference phases.

To evaluate our approach, we introduce the Inference Flow Consistency (IFC) metric, reflecting xFLOW severity. We also utilize Inception Score (IS) (Salimans et al., 2016) and Fréchet Inception Distance (FID) (Heusel et al., 2017) for assessing generation quality. Our models, trained from scratch and compared against baselines on CIFAR-10 (Krizhevsky et al., 2009) and MNIST (LeCun et al., 1998), demonstrate not only an avoidance of xFLOW but also an enhancement in generation quality. The contributions of this paper include:

- We identify a widespread phenomenon in diffusion models, termed xFLOW, leading to non-straight flow during inference that may generate OOD or suboptimal samples.

- We attribute xFLOW's origins to the instability of the target during the training process. Accordingly, we introduce the *Non-Cross Diffusion*, a novel training and inference pipeline to mitigate the xFLOW problem by enhancing input dimensionality.

- Our experiments on both a toy model and image generation on CIFAR-10 and MNIST dataset demonstrate that our method not only improves the proposed IFC metric by addressing xFLOW, but also significantly enhances other image evaluation metrics, such as IS and FID.

## 2 Related work

### 2.1 Diffusion models

Diffusion models, as generative models, learn the reverse denoising process from Gaussian noise to image distribution, achieved through either Markov (Ho et al., 2020) or non-Markov operations (Song et al., 2020a). They are favored over other generative models like GAN (Goodfellow et al., 2014) and VAE (Vahdat & Kautz, 2020) due to their training stability and superior generation quality. Subsequent enhancements to these models primarily concern varied network architectures (Karras et al., 2022), noise schedulers or losses (Nichol & Dhariwal, 2021), transition from image space diffusion to latent space (Rombach et al., 2022), and improved sampling techniques (Lu et al., 2022), with little attention to the xFLOW during training. Rectified Flow (Liu et al., 2022) noted the mismatch in sampling across different inference steps, a significant distillation issue, but did not analyze it further. Instead, they proposed a workaround using a 2-rectified flow to fit another model to a non-crossing flow between source and target distributions, which depends on a well-trained diffusion model and requires additional retraining. Our paper is the first to examine xFLOW in diffusion and offer solutions.

### 2.2 Conditional Image Generation

Conditioning techniques are instrumental in managing generated content. For diffusion models, Song et al. (2020b) propose classifier guidance, which is an efficient method to balance controllability and fidelity using the gradients from a classifier, while classifier-free guidance (Ho & Salimans, 2022), being another important conditioning technique to diffusion models, trains both conditional and unconditional diffusion models, and combining their score to achieve better controllability. ControlNet (Zhang et al., 2023) employs pretrained encoding layers from billions of images as a backbone to learn diverse conditional controls, which is an architecture adopted in this paper. Analog Bits (Chen et al., 2022) introduces a technique that conditions the model on its own previously generated samples during iterative sampling, akin to our work. However, Analog Bits mainly aims to enhance sample quality by reusing the previous target, while our focus is to introduce a condition in the training flow to prevent crossing issues.

## 3 Method

In this section, we start with a brief review of the formulation of DDPM (Ho et al., 2020). Next, we show the drawback of baseline flow and analyze the cause of xFLOW. Then, we introduce *Non-Cross Diffusion* to avoid crossing by ascending dimension of input, together with training, inference, and network architecture of *Non-Cross Diffusion*. Finally, we introduce IFC for evaluating the consistency of the inference flow.

### 3.1 Preliminary

Given samples from data distribution $x_0 \sim q(x_0)$, DDPM (Ho et al., 2020) defines a forward noising process $q$ which produces latent variables $x_1, \ldots, x_T$ by gradually adding Gaussian noise with a variance schedule $\beta_t \in (0, 1)$ as follows:

$$q(x_1, \ldots, x_T) \coloneqq \prod_{t=1}^{T} q(x_t | x_{t-1}), \tag{1}$$

$$q(x_t | x_{t-1}) \coloneqq \mathcal{N}(x_t; \sqrt{1 - \beta_t} x_{t-1}, \beta_t \mathbf{I}). \tag{2}$$

With $\alpha_t \coloneqq 1 - \beta_t$ and $\bar{\alpha}_t \coloneqq \prod_{s=0}^{t} \alpha_s$, the marginal $q(x_t | x_0)$ can be derived through Eq. 2 as follows:

$$q(x_t | x_0) = \mathcal{N}(x_t, \sqrt{\bar{\alpha}_t} x_0, (1 - \bar{\alpha}_t) \mathbf{I}), \tag{3}$$

$$x_t = \sqrt{\bar{\alpha}_t} x_0 + \sqrt{1 - \bar{\alpha}_t} \epsilon, \tag{4}$$

where $\epsilon \sim \mathcal{N}(\mathbf{0}, \mathbf{I})$. Using Bayes theorem, we can calculate the posterior $q(x_{t-1} | x_t, x_0)$ in terms of $\beta_t, \alpha_t$ and $\bar{\alpha}_t$. There are many different ways to parameterize $p_\theta$ to approximate the posterior, while DDPM (Ho et al.,

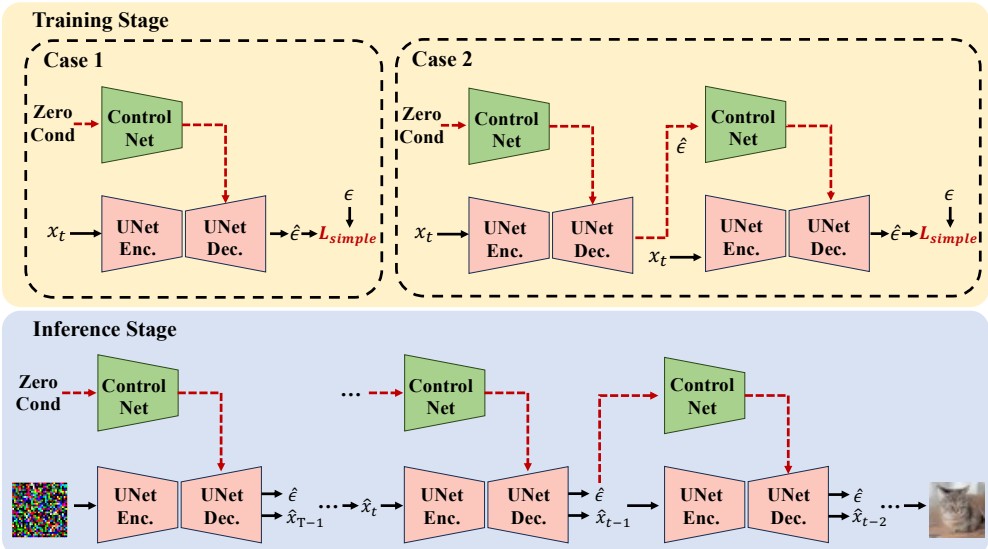

Figure 2: The overview of non-cross diffusion. **Training stage**: The training phase involves two cases. In Case 1, we utilize $\mathbf{0}$ as the condition and calculate loss function $L_{simple}$ as defined in Eq. 5. For Case 2, we first compute $\hat{\epsilon}$ using $\mathbf{0}$ as condition. Subsequently, $\hat{\epsilon}$ is employed as the condition to calculate $L_{simple}$. Throughout the training process, Case 1 is applied with a fixed probability $p$; otherwise, Case 2 is implemented. **Inference stage**: During the inference phase, $\mathbf{0}$ is used as the condition in the initial denoising step. This is followed by iterative utilization of the estimated noise from the previous step as the condition for subsequent steps.

2020) chooses $p_\theta(x_{t-1}|x_t) = \mathcal{N}(x_{t-1}; \boldsymbol{\mu}_\theta(x_t, t), \sigma_t^2 \mathbf{I})$, and propose that predicting $\epsilon$ works best with a loss function:

$$L_{simple} = E_{t,x_0,\epsilon}[\|\epsilon - \epsilon_\theta(x_t, t)\|^2], \tag{5}$$

where $\boldsymbol{\mu}_\theta(x_t, t) = \frac{1}{\sqrt{\alpha_t}}(x_t - \frac{\beta_t}{\sqrt{1-\bar{\alpha}_t}}\epsilon_\theta(x_t, t))$

### 3.2 Understanding Drawbacks of DDPM Flow

**Training stage.** Given source distribution $\pi_0$ (*i.e.*, $q(x_0)$) and target distribution $\pi_1$ (*i.e.*, $\mathcal{N}(\mathbf{0}, \mathbf{I})$), we sample two data pairs $(x_0, x_T), (y_0, y_T) \sim \pi_0 \times \pi_1$. During the training stage, assume these two training flows cross at time step $t$ (*i.e.*, $x_t = y_t$). Following Eq. 4, we have:

$$x_t = \sqrt{\bar{\alpha}_t}x_0 + \sqrt{1-\bar{\alpha}_t}\epsilon_x, \tag{6}$$

$$y_t = \sqrt{\bar{\alpha}_t}y_0 + \sqrt{1-\bar{\alpha}_t}\epsilon_y. \tag{7}$$

At the crossing point, both flows aim to minimize the loss function as follows during training:

$$L_{simple} = E_{t,x_0,\epsilon}[\|\epsilon - \epsilon_\theta(x_t, t)\|^2], \tag{8}$$

For given $(x_0, x_T), (y_0, y_T)$ and cross point $t$, the loss can be reformulated as follows:

$$L_1 = \frac{1}{2}\|\epsilon_x - \epsilon_\theta(x_t, t)\|^2 + \frac{1}{2}\|\epsilon_y - \epsilon_\theta(y_t, t)\|^2, \tag{9}$$

where $\epsilon_x = x_T$ and $\epsilon_y = y_T$. To check how the crossing point affects the optimization process, we simplify the target in the formulation of an optimization problem. Since $\epsilon_\theta(x_t, t) = \epsilon_\theta(y_t, t) = \epsilon_\theta(z_t, t)$, we use the

notion of $\epsilon_\theta$ to represent them all.

$$\theta^* = \arg\min_\theta \frac{1}{2}\|\epsilon_x - \epsilon_\theta\|^2 + \frac{1}{2}\|\epsilon_y - \epsilon_\theta\|^2 \tag{10}$$

$$= \arg\min_\theta \frac{1}{2}(\epsilon_x^2 - 2\epsilon_x\epsilon_\theta + \epsilon_\theta^2 + \epsilon_y^2 - 2\epsilon_y\epsilon_\theta + \epsilon_\theta^2) \tag{11}$$

$$= \arg\min_\theta \frac{1}{2}(\epsilon_x^2 + \epsilon_y^2) - (\epsilon_x + \epsilon_y)\epsilon_\theta + \epsilon_\theta^2 \tag{12}$$

$$= \arg\min_\theta \frac{1}{4}(\epsilon_x^2 + 2\epsilon_x\epsilon_y + \epsilon_y^2) - (\epsilon_x + \epsilon_y)\epsilon_\theta + \epsilon_\theta^2 \tag{13}$$

$$= \arg\min_\theta \|\frac{\epsilon_x + \epsilon_y}{2} - \epsilon_\theta\|^2 \tag{14}$$

Hence, with the existence of a crossing point, the optimizing target is equivalent to $\frac{\epsilon_x+\epsilon_y}{2}$. However, since $\epsilon_x \sim \mathcal{N}(\mathbf{0}, \mathbf{I})$ and $\epsilon_y \sim \mathcal{N}(\mathbf{0}, \mathbf{I})$, we have $\frac{\epsilon_x+\epsilon_y}{2} \sim \mathcal{N}(\mathbf{0}, \frac{\sqrt{2}}{2}\mathbf{I})$, which no longer follows standard normal distribution. This implies that at the crossing point, the model is given *an incorrect target*, which will lead to *ambiguity* in data generation (*i.e.*, the denoising process).

**Inference stage.** Considering the *ambiguity* exists in a trained model, xFLOW, where the generation flow may deviate from the correct direction, results in various failure cases, as illustrated in Fig. 1. Besides, we propose that the consequences of xFLOW also depend on the timestep of inference. Specifically, more inference steps, correlating with smaller strides, subtly affect inference flow because the deviation is also smaller, and subsequent steps can correct minor errors. On the contrary, fewer inference steps, leading to larger strides, significantly impact and alter the inference flow due to the crossing point, potentially generating inconsistent or OOD samples. Such phenomena further decrease the determinism of diffusion models.

### 3.3 Non-Cross Diffusion

As analyzed in Sec. 3.2, xFLOW is caused by incorrect training targets. To solve xFLOW, in this section, we introduce a new formulation of diffusion models that can avoid crossing points during training, namely *Non-Cross Diffusion*.

Given the fact that latent variables are linear combinations of $x_0$ and $\epsilon$ as in Eq. 4. We can think of the issue with geometry, where training flows are line segments in 2D coordinates, as shown in Fig. 1 (a), with the crossing point as the intersection of two segments. From a basic geometrical concept, *i.e.*, any two distinct lines in a plane can intersect at most once, as long as we can avoid the intersection once, the two segments will never intersect again. Therefore, we aim to eliminate crossing points between any two different training flows, thereby maintaining the integrity and distinctiveness for all of them.

To operationalize this concept, we propose to ascend the dimension of model input. The primary issue with Eq. 9 is that $\epsilon_\theta(x_t, t) = \epsilon_\theta(y_t, t)$ when $x_t = y_t$. To rectify this, we can introduce condition $c^x \neq c^y$ to ensure $\epsilon_\theta(x_t, c^x, t) \neq \epsilon_\theta(y_t, c^y, t)$ and prevent the training flow from crossing. The challenge is identifying $c^x \neq c^y$ given a cross point $t$ and training flow. To solve this, we use $x_i, y_i$ as conditions for each non-crossing step $i$ on the training flow, i.e., $c^x = x_i$ and $c^y = y_i$. Specifically, given $x_t = y_t$, by sampling another point on the flow (*i.e.*, $x_i$ and $y_i$), we have $x_i \neq y_i$, and thus $[x_i, x_t] \neq [y_i, y_t], \forall i \in [0, T] \setminus \{t\}$. This reminds us that any other samples $(x_i)$ from the same flow can be used for ascending dimensions. This strategy effectively creates a multidimensional space where the likelihood of training flows intersecting is significantly reduced.

**Selection of Condition.** $x_i$ is effective for ascending dimensions only if it is significantly different from $x_t$. Given the continuity of both linear combination and diffusion models, we propose to ascend the dimension with either initial noise $x_T$ (*i.e.*, $\epsilon$) or the data point $x_0$. Furthermore, we find the distance between randomly sampled noise is stable while the distance between data points may not. Take image data as an example, for two randomly sampled noise $n_1, n_2 \in \mathbb{R}^{H \times W \times C}$, we have $E[\|n_1 - n_2\|^2] = 2CHW$. Besides, we can only get the initial noise during the inference stage. Therefore, using the initial noise $x_T$ for dimension ascending is more practical.

**Training Stage.** The cornerstone of our training strategy is to circumvent trivial solutions and avoid training collapse. To achieve this, we replace the use of initial noise $\epsilon$ with predicted noise $\hat{\epsilon}$. This substitution is critical in refining our model's predictive accuracy since it can effectively avoid trivial solutions. However, in the initial training phase, the substantial error $\|\epsilon - \hat{\epsilon}\|^2$ indicates $\hat{\epsilon}$'s poor estimation. Hence, a bootstrap strategy is introduced to $\hat{\epsilon}$ during training, preventing misleading estimation of $\hat{\epsilon}$ and thus enhancing learning robustness in the early stage.

As illustrated in Fig. 2, our training objective is formulated as follows:

$$\min_\theta E_{t,x_t,\epsilon,}[\|\epsilon - \epsilon_\theta(x_t, \hat{\epsilon}_t, t)\|^2], \tag{15}$$

where $x_t = \sqrt{\bar{\alpha}_t}x_0 + \sqrt{1 - \bar{\alpha}_t}\epsilon$. During training, we apply the bootstrap as follows: 1) with a fixed probability $p$, we set $\hat{\epsilon}_t = \mathbf{0}$ (*i.e.*, Case 1 in Fig. 2); 2) at other cases, $\hat{\epsilon}_t$ is assigned the value of $\epsilon_\theta(x_t, \mathbf{0}, t)$ (*i.e.*, Case 2 in Fig. 2). We do not back-propagate through estimated noise $\hat{\epsilon}_t$.

**Inference Stage.** As illustrated in Fig. 2, during the inference stage, to alleviate the computational costs, we use estimated noise in the previous step instead of the current step as the condition and iteratively predict $\hat{\epsilon}$ as follows:

$$\hat{\epsilon}_T = \epsilon_\theta(\hat{x}_T, \mathbf{0}, T), \tag{16}$$

$$\hat{\epsilon}_t = \epsilon_\theta(\hat{x}_t, \hat{\epsilon}_{t+1}, t), t < T. \tag{17}$$

When the number of inference steps is large, the discrepancy between $\hat{\epsilon}_t$ and $\hat{\epsilon}_{t+1}$ is small, which ensures the performance of our method.

**Network Architecture.** Inspired by ControlNet (Zhang et al., 2023), to efficiently use $\hat{\epsilon}_t$, *Non-Cross Diffusion* employs an additive U-net branch, with $\hat{\epsilon}_t$ as input. For optimization, modifications are introduced, specifically removing all zero convolution layers and initializing the addictive encoder for $\hat{\epsilon}_t$ with the original U-net. The output is incorporated into the U-net decoder via addition. The whole network is trained end-to-end from scratch.

### 3.4 Inference Flow Consistency

To better evaluate the consistency of the inference flow for image generation, we propose a metric by computing the similarity between intermediate generated image $\hat{x}_0^t$ in timestep $t$ and the final generated image $\hat{x}_0$ based on peak signal-to-noise ratio (PSNR) as follows:

$$\text{IFC} = \frac{1}{T}\sum_{t=0}^{T} PSNR(\hat{x}_0^t, \hat{x}_0). \tag{18}$$

A change in training flow direction at a specific timestep yields notable differences in pre- and post-change images, reducing PSNR. This can be effectively assessed for consistency across inference stages using our PSNR-based metric.

## 4 Experiment

In this section, we discuss our experimental results on toy examples (Sec. 4.1) and image generation tasks (Sec. 4.2), as well as ablation studies (Sec. 4.3) and further discussion for *Non-Cross Diffusion* (Sec. 4.4).

### 4.1 Toy Examples

In this section, we follow the setting in Rectified Flow (Liu et al., 2022), drawing a training dataset from Gaussian mixture $\pi_0 \times \pi_1$. Given a sample $\{x_0^i, x_1^i\}$ from $(X_0, X_1) \sim \pi_0 \times \pi_1$, for baseline model, we train a 3-layer MLP $v_\theta(z, t)$ to transfer from $\pi_0$ to $\pi_1$ with l2-loss as follow:

$$\min_\theta \|v_\theta(x_t^i, t) - (x_1^i - x_0^i)\|_2,$$

$$x_t^i = tx_1^i + (1 - t)x_0^i, t \in [0, 1).$$

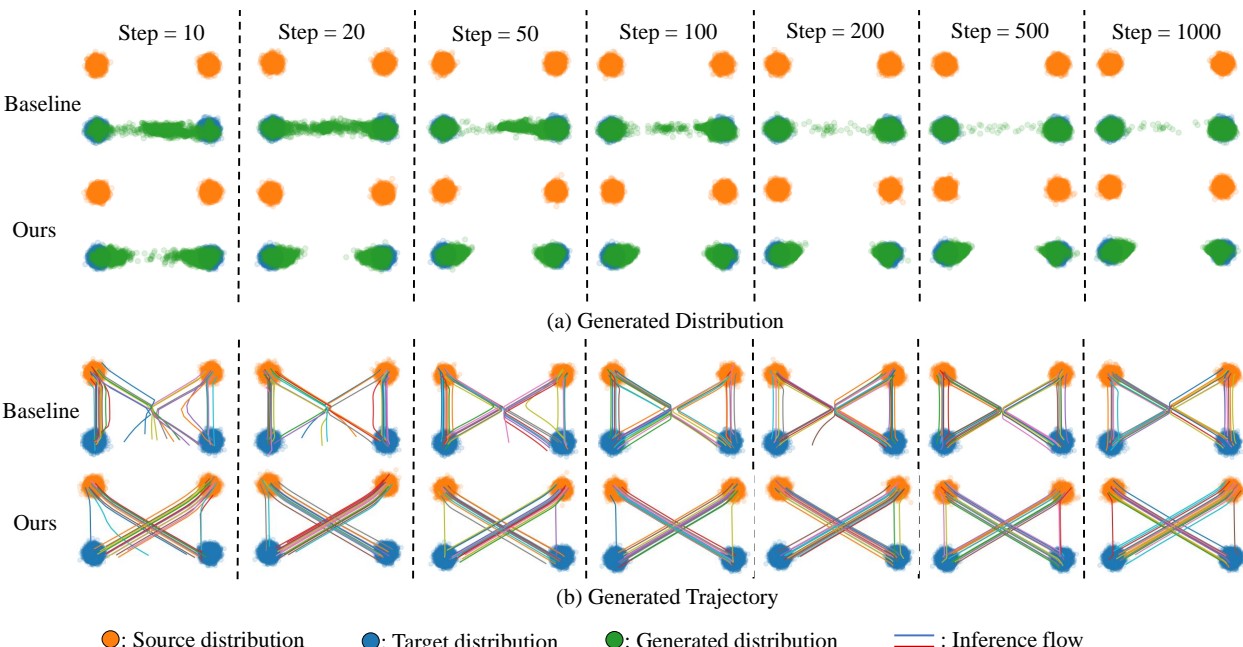

(a) Generated Distribution

(b) Generated Trajectory

●: Source distribution    ●: Target distribution    ●: Generated distribution    ——: Inference flow

Figure 3: Results of the Toy Model. (a) Comparison of Generated Distributions: This panel illustrates the distributions generated by the baseline model and our proposed model. As the number of inference steps decreases, the baseline model tends to produce a significant number of out-of-distribution (OOD) samples. In contrast, our model effectively mitigates the generation of OOD samples. (b) Trajectory Analysis: This panel compares the generated trajectories of the baseline and our models. The baseline model's inference flow often redirects at the intersection point, leading to a target OOD distribution as the inference steps decrease. Our method, however, maintains a consistent direction in the inference model, thereby straightening the trajectory.

| DDIM-1000 | | | DDIM-100 | | | DDIM-50 | | |
|---|---|---|---|---|---|---|---|---|
| Method | IS | FID | Method | IS | FID | Method | IS | FID |
| iDDPM | 9.02 | 4.70 | iDDPM | 8.99 | 5.65 | iDDPM | 8.89 | 6.61 |
| iDDPM‡ | 9.10 | 4.82 | iDDPM‡ | 8.93 | 5.75 | iDDPM‡ | 8.79 | 6.71 |
| Ours | **9.51** | **2.88** | Ours | **9.22** | **3.93** | Ours | **9.10** | 5.31 |
| Ours† | 9.34 | 3.40 | Ours† | 9.15 | 4.21 | Ours† | 9.05 | **5.09** |
| DDIM-20 | | | DDIM-10 | | | DDIM-5 | | |
| Method | IS | FID | Method | IS | FID | Method | IS | FID |
| iDDPM | 8.59 | 9.82 | iDDPM | 8.20 | 15.91 | iDDPM | 7.09 | 31.37 |
| iDDPM‡ | 8.65 | 9.89 | iDDPM‡ | 8.14 | 16.06 | iDDPM‡ | 7.08 | 31.21 |
| Ours | 8.77 | 9.87 | Ours | 7.97 | 20.63 | Ours | 6.20 | 50.25 |
| Ours† | **8.84** | **7.75** | Ours† | **8.50** | **12.85** | Ours† | **7.45** | **27.83** |

Table 1: We compare the performance of baseline and our method. We generate 50k samples using DDIM with inference steps in {1000, 100, 50, 20, 10, 5}. ‡We expand the U-net encoder to ensure the same model size as ours. †We use an inference strategy similar to the training stage. Specifically, we first give **0** as condition and get estimated noise $\hat{\epsilon}_t$, then we take $\hat{\epsilon}_t$ as condition and compute denoised image $\hat{x}_{t-1}$.

Our method enhances this approach by incorporating an additional dimension with the estimated target as follows:

$$\min_{\theta} \|v_\theta([x_t^i, \hat{c}_t^i], t) - (x_1^i - x_0^i)\|_2,$$

$$\hat{c}_t^i = \begin{cases} \mathbf{0} & \text{p} \leq 0.5 \\ v_\theta([x_t^i, \mathbf{0}], t) & \text{otherwise} \end{cases}$$

| DDIM-1000 | | DDIM-100 | | DDIM-50 | |
|---|---|---|---|---|---|
| Method | FID | Method | FID | Method | FID |
| iDDPM | 8.02 | iDDPM | 8.26 | iDDPM | 9.12 |
| Ours | **7.13** | Ours | **7.72** | Ours | 9.06 |
| Ours* | 7.52 | Ours* | 7.91 | Ours* | **8.76** |

Table 2: We compare the performance of baseline and our method on MNIST. We generate 50k samples using DDIM with inference steps in {1000, 100, 50}. †We use an inference strategy similar to the training stage.

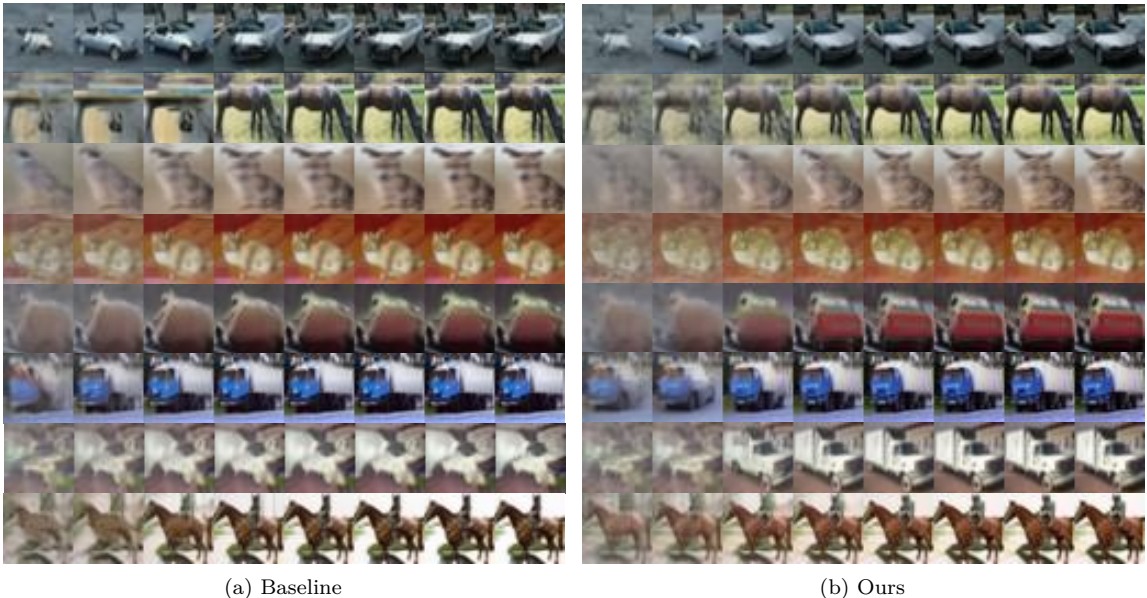

(a) Baseline                    (b) Ours

Figure 4: Here are the generated images using DDIM with inference steps in {5, 10, 25, 50, 100, 200, 500, 1000} on CIFAR-10. For baseline method, the semantic information of image with small inference step and large inference step could be greatly different, which implies that the inference flow changes its direction at some timesteps.

with $p \sim U(0, 1)$. The inference process is also similar to our proposed method, *i.e.*, we use the estimated target in the previous step as the condition.

**Results.** As shown in Fig. 3, the baseline model's inference flow alters direction at the intersection of two flows due to an erroneous loss function (Eq. 14), generating OOD samples. In Sec. D, we further discuss the intersection and direction altering in real life model and datasets. For toy examples, by adding an extra dimension using the estimated result, our method prevents training and inference flow intersection, maintaining consistent inference flow direction and effectively inhibiting OOD sample generation.

## 4.2 Experiments on Image Generation

**Implementation Details.** Our models are trained on CIFAR-10 (Krizhevsky et al., 2009) and MNIST (Le-Cun et al., 1998), with MNIST images resized to $32 \times 32$. The fidelity of generated samples is evaluated using IS (Salimans et al., 2016) and FID (Heusel et al., 2017) and inference flow consistency with IFC. As a baseline, we train iDDPM (Nichol & Dhariwal, 2021) from scratch with the same UNet. Training for CIFAR-10 follows iDDPM except using $L_{simple}$ only and with 250k steps. For MNIST, training is similar to CIFAR-10 but with 100k steps. In this paper, we consider unconditional generations on each dataset (*i.e.*, w/o class labels).

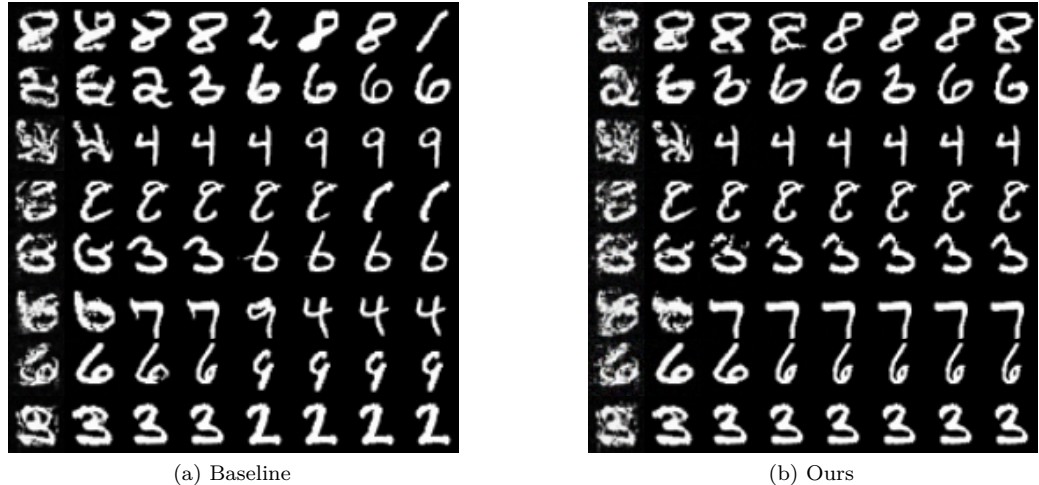

(a) Baseline       (b) Ours

Figure 5: Here are the generated images using DDIM with inference steps in {5, 10, 25, 50, 100, 200, 500, 1000} on MNIST.

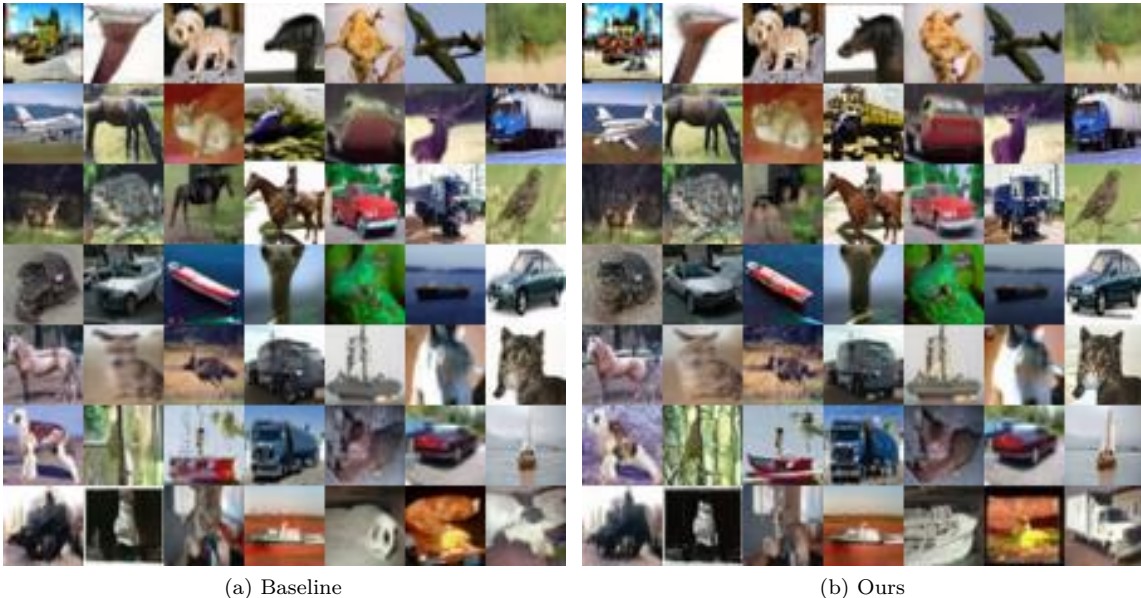

(a) Baseline       (b) Ours

Figure 6: Displayed are the generated images of baseline model and our model using DDIM with inference step of 1000 on CIFAR-10. The results demonstrate our model's superior image generation capabilities, significantly reducing the occurrence of OOD samples.

**Comparison of Sampling Quality.** Tables 1 and 2 compare our model's performance with a baseline model in generating CIFAR-10 and MNIST images. Fig. 6 visualizes the generated CIFAR-10 images, where our model notably outperforms the baseline, especially at {1000, 100, 50} inference steps. The decreased number of inference steps and increased strides enlarge the discrepancy in estimated noise between steps, introducing bias and performance decline during inference. To counter this, we suggest conditioning on the current step's estimated noise. This augments image quality generated by our method even at smaller steps in {20, 10, 5}. Besides, our model outperforms the baseline on MNIST at steps {1000, 100, 50}. The adapted sampling strategy also enhances our model's performance on MNIST as the number of inference steps decreases.

| DDIM-1000 | | DDIM-100 | | DDIM-50 | |
|---|---|---|---|---|---|
| Method | IFC | Method | IFC | Method | IFC |
| iDDPM | **28.58** | iDDPM | 29.72 | iDDPM | 30.85 |
| Ours | 28.37 | Ours | **29.96** | Ours | **31.40** |
| DDIM-20 | | DDIM-10 | | DDIM-5 | |
| Method | IFC | Method | IFC | Method | IFC |
| iDDPM | 33.94 | iDDPM | 38.51 | iDDPM | 46.11 |
| Ours | **35.11** | Ours | **40.21** | Ours | **47.96** |

Table 3: We compare the consistency of baseline and our method on CIFAR-10. We generate 1000 samples using DDIM with inference steps in {1000, 100, 50, 20, 10, 5}.

| | Setting | IS | FID |
|---|---|---|---|
| Bootstrap | w/o bootstrap | 5.93 | 79.70 |
| | exp-schedule | 9.22 | 5.24 |
| | fix-prob (Ours) | **9.38** | **4.84** |
| Condition | $\hat{x}_0$ condition | **9.64** | 20.21 |
| | mid. condition | 7.15 | 59.09 |
| | $\hat{\epsilon}$ condition (Ours) | 9.38 | **4.84** |
| Network | Double Unet | **9.65** | 6.26 |
| | ControlNet-based (Ours) | 9.38 | **4.84** |
| Inference strategy | zero condition | 9.06 | 6.43 |
| | $\epsilon$ condition | 7.86 | 27.00 |
| | $\hat{\epsilon}$ condition (Ours) | **9.38** | **4.84** |

Table 4: Ablation study. We generate 10k samples using DDIM with 1000 inference step.

**Comparison of Inference Consistency.** Table 3 reveals our method's superior consistency over the baseline on CIFAR-10 at inference steps {50, 20, 10, 5}, in terms of IFC, achieved by preventing xFLOW. The impact of xFLOW is minimal for larger steps ({1000, 100}), resulting in similar IFC between our method and the baseline. Fig. 4 and Fig. 5 illustrate the visualization results of generated images under different steps on CIFAR-10 and MNIST. The results further demonstrate higher consistency compared with baselines, indicating a straighter inference flow.

### 4.3 Ablation Study

In this section, we verify the effectiveness of each component through ablation studies.

**Ablation of bootstrap.** Table 4 demonstrates the impact of the bootstrap strategy. We experimented with three settings: without bootstrap (w/o bootstrap, where $\hat{\epsilon}_t = \epsilon_\theta(x_t, \mathbf{0}, t)$ is consistently applied), exponential schedule (exp-schedule, where the probability $p$ increases exponentially), and fixed probability (fix-probability, where $p = 0.5$). The findings reveal that lacking a bootstrap strategy can notably degrade model performance.

**Ablation of condition.** We scrutinize the impact of varying conditions on our method. The $\hat{x}_0$ condition implies the model utilizes the predicted image as the condition, while the mid. condition uses the midpoint of training flow (*i.e.* $\frac{\hat{x}_0 + \hat{\epsilon}}{2}$). During inference, each technique employs its corresponding condition. As in Table 4, mid. performs suboptimally, presumably due to proximity to the training flows' intersection, as mentioned in Sec. 3.3. Benefiting from its stability, $\hat{\epsilon}$ as a condition leads to a significant FID improvement over $\hat{x}_0$ condition.

**Ablation of architecture.** We also examine a Double U-net variant, *i.e.*, doubling the U-net's input channel to accommodate $\hat{\epsilon}$ input. As Table 4 shows, the ControlNet-based model enhances FID by 1.42, likely because the two inputs serve distinct roles, and the additive branch enables effective differentiation between them, thereby facilitating training.

**Ablation of inference strategy.** We also consider several inference strategies applied after training (we utilize a trained $\hat{\epsilon}$-conditioned model). These inference strategies include the Zero condition (utilizing $\mathbf{0}$), the $\epsilon$ condition (using initial noise $\epsilon$), and the $\hat{\epsilon}$ condition (employing estimated noise $\hat{\epsilon}$). Performance significantly deteriorates under the $\epsilon$ condition due to $\epsilon$-$\hat{\epsilon}$ discrepancy. Though our model circumvents training ambiguity, the Zero condition marginally compromises performance during inference due to potential redirection at cross points in the inference flow.

### 4.4 Discussion

This section offers an alternative perspective to understand our *Non-Cross Diffusion*. The inclusion of $\hat{\epsilon}_t$ in the model input is seen as a strong conditional constraint, which we propose can reduce the variability of semantic information, thus lessening the severity of XFLOW during inference.

Specifically, the $\hat{\epsilon}_t$ condition in *Non-Cross Diffusion* is highly specific at the pixel level, making it an exceptionally stringent constraint. Consequently, *Non-Cross Diffusion* effectively mitigates the XFLOW issue. An interesting question arises: how would XFLOW be influenced by other forms of conditions with varying degrees of strength? To investigate this, we employed the pre-trained ControlNet model for empirical analysis and results.

Fig. 1(c) shows text conditional images from Stable Diffusion (Rombach et al., 2022), as well as pose conditional and depth conditional images from ControlNet (Zhang et al., 2023). The text condition images at steps $\{5, 10, 25, 50, 250, 500\}$ present a significant shift in the semantic content of the images at each step. In contrast, pose conditional images demonstrate more consistent semantic content across steps: the pose of super-hero remains the same but the background and style still show a large variation. The depth conditional images keep the highest consistency across different steps, despite some variability in details such as the background and pattern. Therefore, we hypothesize that stronger conditions ease the severity of XFLOW.

## 5 Conclusion

In this study, we address the XFLOW phenomenon in diffusion models, characterized by deviations in generative flow that result in semantic inconsistencies and suboptimal image generation. Our novel approach, '*Non-Cross Diffusion*', innovates in the realm of generative modeling by adopting ordinary differential equation models. Our empirical investigations, including both a toy example and image dataset such as CIFAR-10 and MNIST, demonstrate the substantial efficacy of the *Non-Cross Diffusion* approach. The results show a marked reduction in semantic inconsistencies at various inference stages and significant improvements in the overall performance of diffusion models.

**Looking Ahead.** The identification of XFLOW as a critical issue during inference opens new avenues for research and application optimization. Despite the effectiveness of the proposed *Non-Cross Diffusion* approach on mitigating XFLOW, we acknowledge the challenges associated with retraining large-scale diffusion models such as Stable Diffusion. However, we are optimistic that future research will find ways to integrate these improvements into existing models, potentially circumventing the need for extensive retraining. This paper lays the groundwork for such advancements, aiming to enhance the reliability and quality of diffusion model outputs.

**Broader Impact Statements**

This paper delineates our pivotal contribution towards fostering advancement in diffusion models within the purview of generative models. The societal implications of our research are multitudinous and profound. Some of them have been presented in Sec. 1. Additionally, the eminent stable diffusion series has been discussed as a case study to illustrate potential applications of our work in Sec. 4.4. Furthermore, we delve into the potential impact of our research from the standpoint of fellow academicians in Sec. 5, where we also deliberate on future directions for this line of inquiry. There are no additional aspects we deem imperative to underscore at this juncture.

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

# APPENDIX

## A  Continuity of Diffusion Models

In Sec. 3.3, we select the condition based on the inherent continuity in diffusion models. Continuity, although abstract, is pivotal to understanding why we select noise as the condition. To make this concept more tangible, we demonstrate it by introducing variable scale perturbations at different time steps. Specifically, for a fixed timestep $t$ and diffusion model $\epsilon_\theta$, the $\epsilon_\theta(x_t, t)$ is continuous with respect to $x_t$. This implies that for any $\tilde{x}_t$ close to $x_t$, the resulting image remains similar, which shows the continuity principle in diffusion models.

### A.1  Perturbation on Noise

Practically, for initial noise $\epsilon$, *i.e.* $t = T$, we add perturbation as follows:

$$\tilde{\epsilon} = \frac{\epsilon + w \cdot \epsilon_p}{\sqrt{1 + w^2}}, \tag{19}$$

where $\epsilon_p$ is a perturbation sampled from a standard Gaussian distribution and $w$ denotes the scale of perturbation. For intermediate $x_t$ in timestep $t$, we first estimate the noise through the baseline unconditional diffusion model:

$$\hat{\epsilon} = \epsilon_\theta(x_t, t). \tag{20}$$

Then we add perturbation to the estimated noise $\hat{\epsilon}$:

$$\tilde{\epsilon} = \frac{\hat{\epsilon} + w \cdot \epsilon_p}{\sqrt{1 + w^2}}, \tag{21}$$

and we get the denoised image as follows:

$$x_{t-1} = \sqrt{\bar{\alpha}_{t-1}}\hat{x}_0 + \sqrt{1 - \bar{\alpha}_{t-1}}\tilde{\epsilon}, \tag{22}$$

where $\hat{x_0}$ is predicted by $\hat{\epsilon}$ and $x_t$.

### A.2  Result with Perturbation

As depicted in Fig. 8, with a minimal scale, such as 0.01, there is negligible impact on the semantic content of the images across different steps. The semantic information of generated images remains intact. As the scale increases, the disparity between the original and the perturbed images becomes more pronounced. At a scale of 0.2, the semantic alterations are most notable, as exemplified by significant changes in elements like the postures of animals in the images. For instance, the positions of a cat or a deer might shift noticeably. Despite these changes, the overall semantic context — the fundamental identity of the subjects — remains unchanged, underscoring the continuity intrinsic to diffusion models.

However, at a higher scale, such as 0.5, the semantic shift becomes dramatic. The generated images undergo substantial transformations, to the extent that the class of the subject in the image can completely change, such as transitioning from a `cat` to a `frog`.

The observation shows that for the cross point, *i.e.* the noised images are identical, with similar conditions, the generated image will still remain the same and cannot avoid crossing. With more distinct conditions, the effectiveness of avoiding crossing increases.

## B  Details of our model

**ControlNet-based.** Fig. 7 presents the architecture of our proposed model. Notably, we have omitted the zero convolution layer originally found in the ControlNet (Zhang et al., 2023) design. Both the UNet and Control Blocks are concurrently trained from scratch. The outputs from Control Blocks A, B, C, and D are

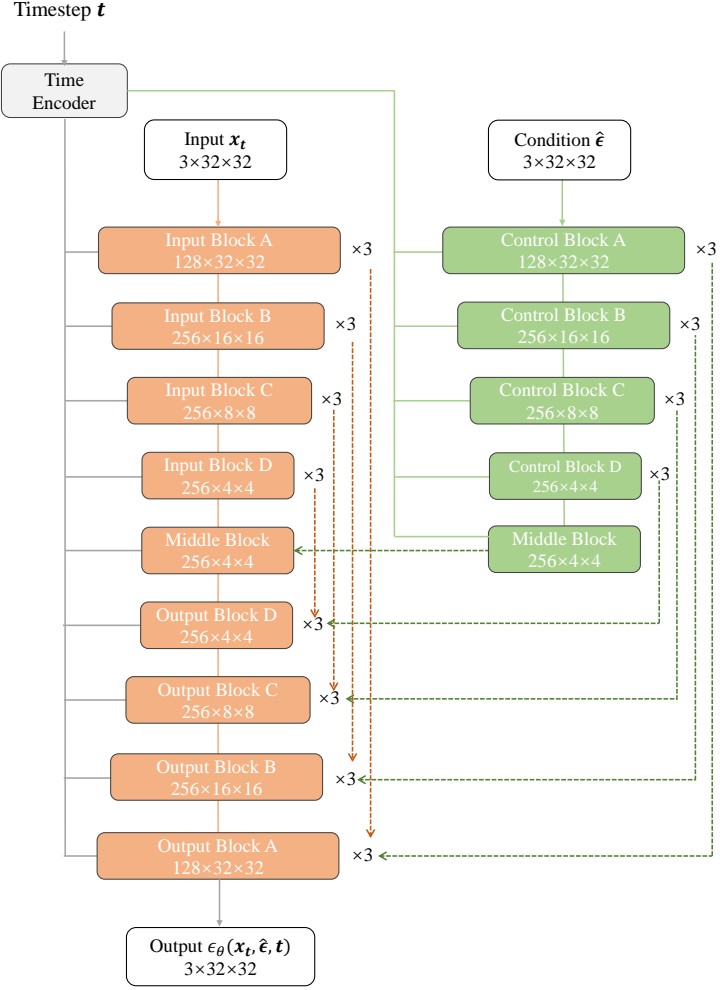

Figure 7: The architecture of our model. We show the output shape of each module.

integrated with the respective outputs of the input blocks. This integration is achieved by adding the outputs from the Control Blocks to those of the Input Blocks, followed by concatenation with the corresponding output from the Output Blocks. The output from the middle block of ControlNet is directly added to the output of UNet's middle block, and then this combined output is fed into UNet's output block D.

**Double UNet in Ablation Study.** For the Double UNet setting in the ablation study, we modified the input channel of the first layer in UNet. This modification involved doubling the input channel, *i.e.*, changing the input channel of the first convolution layer from 3 to 6.

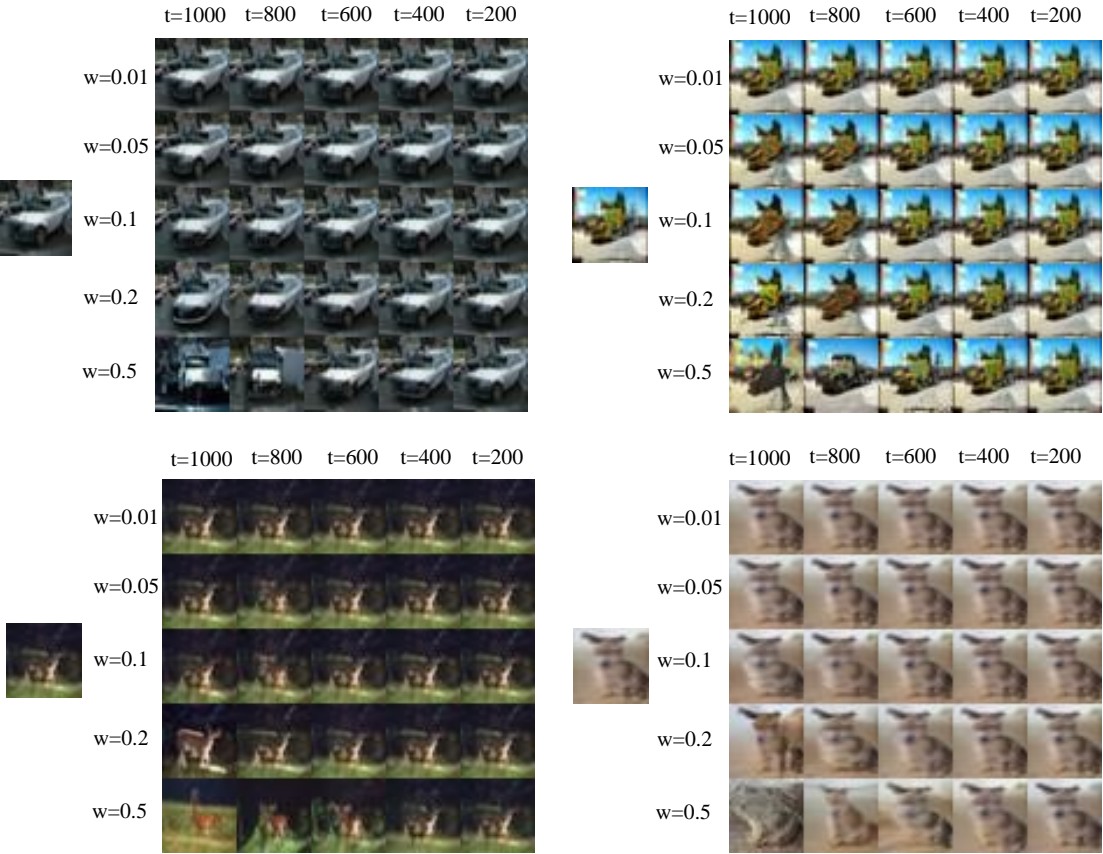

Figure 8: We sample images using iDDPM (Nichol & Dhariwal, 2021) with 1000 inference steps on CIFAR-10. For the image grid, from left to right, the perturbation is added at $t = 1000, 800, 600, 400, 200$. From top to bottom, the perturbation is added with weight $w = 0.01, 0.05, 0.1, 0.2, 0.5$. We sample all images with DDIM (Song et al., 2020a)

## C    Training and Inference Details

**Trivial solutions with initial noise as a condition.** In the training stage of the diffusion model, the loss function is:

$$L_{simple} = E_{t,x_0,\epsilon}[\|\epsilon - \epsilon_\theta(x_t, t)\|^2], \tag{23}$$

If we use initial noise as a condition, then the loss function will be:

$$L_{simple} = E_{t,x_0,\epsilon}[\|\epsilon - \epsilon_\theta(x_t, \epsilon, t)\|^2], \tag{24}$$

Therefore, during training, since we use the target $\epsilon$ as input, the model $\epsilon_\theta(x_t, \epsilon, t)$ can directly output $\epsilon$ without learning anything. We call such a model the trivial solution from diffusion loss. Practically, we run an experiment with the settings above. As a result, the $L_{simple}$ will quickly descend to near 0, however, the generated images are almost noise without any useful semantic information. which denotes that using initial noise as a condition leads to trivial solution.

On the contrary, by using predicted noise as a condition, the loss function is:

$$L_{simple} = E_{t,x_0,\epsilon}[\|\epsilon - \epsilon_\theta(x_t, \hat{\epsilon}, t)\|^2], \tag{25}$$

$$\hat{\epsilon} = \epsilon_\theta(x_t, \mathbf{0}, t). \tag{26}$$

Since there is no $\epsilon$ as input of the diffusion model, we can avoid a trivial solution and successfully train diffusion models for generation.

**Training and Inference Algorithm** Algorithm 1 shows the training pipeline of our method. As described in Section 3.3, we replace the use of initial noise $\epsilon$ with predicted noise $\hat{\epsilon}$ to avoid trivial solutions and training collapse. During training, we set $\hat{\epsilon} = \mathbf{0}$, with a fixed probability $p = 0.5$, and otherwise, $\hat{\epsilon}$ is assigned the value of $\epsilon_\theta(x_t, \mathbf{0}, t)$.

Algorithm 2 shows the inference pipeline of our method. We use estimated noise in the previous step instead of the current step as the condition and iteratively predict $\hat{\epsilon}$, which can effectivly alleviate the computational costs.

---

**Algorithm 1** Training Stage

---

**Input:** diffusion model $\epsilon_\theta$, training step $T$
1: **repeat**
2:     sample $x_0 \sim q(x_0), \epsilon \sim \mathcal{N}(0, \mathbf{I}), t \sim \mathcal{U}(0, T-1)$
3:     compute noised image $x_t = \sqrt{\bar{\alpha}_t}x_0 + \sqrt{1 - \bar{\alpha}_t}\epsilon$
4:     sample $p \sim U(0, 1)$
5:     $\hat{\epsilon} \leftarrow \mathbf{0}$
6:     **if** $p \geq 0.5$ **then**
7:         $\hat{\epsilon}, \hat{x}_0 \leftarrow \epsilon_\theta(x_t, \hat{\epsilon}, t)$
8:         $\hat{\epsilon} \leftarrow stop\_gradient(\hat{\epsilon})$
9:     **end if**
10:    $\hat{\epsilon}, \hat{x}_0 \leftarrow \epsilon_\theta(x_t, \hat{\epsilon}, t)$
11:    $Loss = MSE(\epsilon, \hat{\epsilon})$
12: **until** converged

---

**Algorithm 2** Inference Stage

---

**Input:** diffusion model $\epsilon_\theta$, inference step $T$
1: sample $x \sim \mathcal{N}(0, \mathbf{I})$
2: $\hat{\epsilon} \leftarrow \mathbf{0}$
3: **for** $t$ in $range(T, 0)$ **do**
4:     $\hat{\epsilon}, \hat{x}_0 \leftarrow \epsilon_\theta(x, \hat{\epsilon}, t)$
5:     $x = \sqrt{\bar{\alpha}_{t-1}}\hat{x}_0 + \sqrt{1 - \bar{\alpha}_{t-1}}\hat{\epsilon}$
6: **end for**
7: **return** generated image $x$

---

**Exp-schedule Bootstrap in Ablation Study.** For the exp-schedule setting in the ablation study, we modified the probability $p$ of applying condition during the training stage in Alogrithm 1. Specifically, we set $p$ following:

$$p = 0.999^{\frac{step}{100}}, \tag{27}$$

where *step* denotes the training step.

# D    xFlow in Real Life Models

In this section, we further show the xFLOW in high-dimensional space and real-life models (*i.e.*, stable diffusion).

**Existence of Crossing in High-Dimensional Space.** To show the existence of crossing during the training stage on image data, first, we formulate the problem as the following existence preposition:

**Proposition D.1** (existence of crossing point). *Given images or latents $x_1, x_2$, randomly sample noise $n_1 \sim \mathcal{N}(\mathbf{0}, \mathbf{I})$, there exists timestep $t \in [0, 1]$ and noise $n_2 \sim \mathcal{N}(\mathbf{0}, \mathbf{I})$, so that $x_1^t = x_2^t$, where $x_1^t = tx_1 + (1-t)n_1$ and $x_2^t = tx_2 + (1-t)n_2$.*

Since in the training stage of the diffusion model, the timestep is sampled from a fixed set $\mathcal{S}$, like $\{\frac{1}{n}, \frac{2}{n}, ..., \frac{n-1}{n}\}$, the above proposition can be reformulated as follows:

**Proposition D.2** (existence of crossing point with discrete timesteps). *Given images or latents $x_1, x_2$, randomly sample noise $n_1 \sim \mathcal{N}(\mathbf{0}, \mathbf{I})$, there exists timestep $t \in \mathcal{S}$, so that $n_2 = n_1 + \frac{t}{1-t}(x_1 - x_2)$ and $n_2 \in \mathcal{N}(\mathbf{0}, \mathbf{I})$.*

Note that $x_1, x_2$ are given constant vectors, therefore, the term $\frac{t}{1-t}(x_1 - x_2)$ does not change the covariance of noise $n_1$. Therefore, we only need to prove that all the elements of $n_2$ follow $\mathcal{N}(0, 1)$. However, it's still

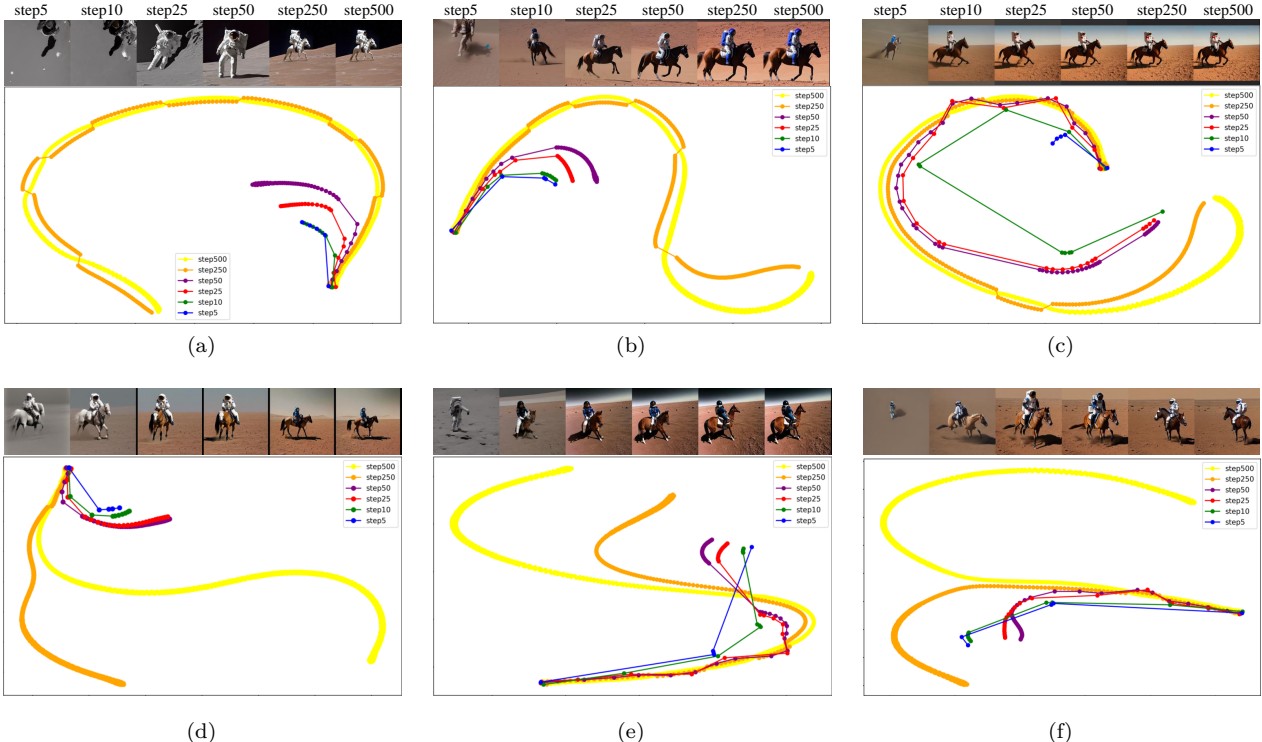

Figure 9: Generated images and inference flows of Stable Diffusion 1.5 using DDIM scheduler with prompt *"a photo of an astronaut riding a horse on mars"* and negative prompt *"bad, deformed, ugly, bad anotomy"*. (a)-(f) are generated with seed 0,1,2,3,4,5 respecitvely. The results demonstrate that the inference flows across different steps could be quite different at specific time $t$, which implies the influence of the xFLOW.

hard to formally prove Proposition D.2, since the distribution of $n_2$ is highly related to the distribution of dataset. Therefore, we empirically verify the existence of crossing in a specific dataset using the Shapiro-Wilk test.

Specifically, we sample 50k image pairs $(x_1, x_2)$ in Cifar-10 and noise $n_1 \sim \mathcal{N}(\mathbf{0}, \mathbf{I})$, and traverse timesteps $t$ in $\mathcal{S} = \{\frac{999}{1000}, \frac{998}{1000}, ..., \frac{100}{1000}\}$ to compute $n_2$. For Shapiro-Wilk test, we set alpha level $\alpha = 0.05$. Therefore, if $n_2$ satisfies the following three requirements, we assume that $n_2$ follows standard normal distribution: (1) the p-value is greater than 0.05; (2) the mean is in $[0 - E_m, 0 + E_m]$; (3) the standard deviation is in $[1 - E_s, 1 + E_s]$. We set $E_m = 0.015$, $E_s = 0.01$ according to the average absolute error of the mean and the standard deviation of randomly sampled data with the same scale.

The experiment shows that the existence ratio of crossing in CIFAR-10 is 19.64%, which implies that even in high-dimensional space, there still exists crossing in the training stage.

**Existence of Redirection in Stable Diffusion.** We further visualize the inference flow of text-to-image Stable Diffusion in different steps. We save the intermediate latent of each step and visualize the inference flow by using T-SNE. As illustrated in Fig. 9, the visualization demonstrates that Stable Diffusion also shows various inference flows across different steps, which implies the influence of xFLOW. The results in Fig. 9 demonstrate that the xFLOW phenomenon occurs frequently within stable diffusion, which results in sub-optimal or even OOD synthesis.

