# OpenReview forum: "Non-Cross Diffusion for Semantic Consistency"
_TMLR — Rejected by TMLR_

### Review · Reviewer_XbZx · 2024-05-07

**Summary Of Contributions:**

This paper first claims that a phenomenon called xFlow can be found in current diffusion models. And then proposed a new method to address the problem. The paper includes detailed comparisons with baseline models, and also did ablation experiments to show the importance of different components of the algorithm.

**Audience:**

Yes

**Claims And Evidence:**

No

**Requested Changes:**

A more detailed analysis of the xFlow phenomenon in real life models would be needed. In two dimension or three dimension, it's quite easy to understand, but when dealing with very high dimensional data, it's difficult to imagine that there could be crossings.

**Strengths And Weaknesses:**

## Strengths

* Comparison with baseline used the same size model. This reduces the likelihood that the proposed model performed better simply because the new models are larger.

* The paper did the toy example experiment, which is quite easy to understand.

## Weaknesses

* It's not clear if the xFlow phenomenon appears in real life models. Because real life models deal with very high dimensional data and the curse of dimensionality, it's difficult to imagine that there could be real "crosses" in the flow path.

---

> ### Author Response · Authors · 2024-05-26
>
> ##    Weaknesses
> **Q1: It's not clear if the xFlow phenomenon appears in real life models. Because real life models deal with very high dimensional data and the curse of dimensionality, it's difficult to imagine that there could be real "crosses" in the flow path.**
>
> A: We further dicuss the xflow in real life model in term of existence of crossing and redirection with different inference step. Please refer to **Comment on Common Question Q1** for details.
>
> ##    Requested Changes:
> **Q1:A more detailed analysis of the xFlow phenomenon in real life models would be needed. In two dimension or three dimension, it's quite easy to understand, but when dealing with very high dimensional data, it's difficult to imagine that there could be crossings.**
>
> A: We further dicuss the xflow in real life model in term of existence of crossing and redirection with different inference step. We have add Sec. D to the Appendix to futher discuss it.

---

### Review · Reviewer_gPCs · 2024-05-09

**Summary Of Contributions:**

This work identifies a phenomena in diffusion model training which is believed to lead to suboptimal sample quality: sample paths with different endpoints ${(x_0, x_T), (x_0', x_T')}$ may cross each other.  The authors argue that this would lead to the model being provided with an ambiguous target, which is especially undesirable if the endpoints of the two trajectories correspond to data points from different classes.  They propose to augment the data samples with the class label to reduce such crossing.  The method is found to outperform the standard training process in terms of sample quality and consistency during training.

**Audience:**

No

**Claims And Evidence:**

No

**Requested Changes:**

Regarding (1) above, it would be helpful to demonstrate if exact path crossing happens in higher-dimensional problems through theoretical and numerical examples.  I suggest having $d\ge 20$ dimensions with a similar scale of variance, in light of previous findings on the intrinsic dimensionality of image data (Pope et al, 2021).

Clarification on (2) above would be welcome.

References:

Pope et al, The Intrinsic Dimension of Images and Its Impact on Learning, ICLR 2021.

**Strengths And Weaknesses:**

Strengths:

- The proposed method is simple and demonstrates improvement over the standard training process.


Weaknesses:

**1.** I have doubts about the soundness of the motivation: generally speaking, the probability of two randomly sampled continuous paths crossing each other should quickly vanish as the dimensionality of the data space increases.  This seems analogous to the fact that two random walks on $\mathbb{Z}^d$ will meet each other with probability 1 if $d\le 2$, and 0 if $d>2$.  Therefore, it's quite unclear if the path crossing phenomenon should ever happen on higher-dimensional datasets such as image datasets.

The authors demonstrated that the crossing phenomenon occurs on a synthetic example with $d=2$.  In light of the above issue, I don't feel it could provide much justification.

It might be argued that more general phenomenon than two paths exactly crossing each other could occur in higher-dimensional problems; for example, while two paths might not cross each other, their projections to certain "principal subspaces" may still do.  But it's quite unclear how to frame this formally and whether such phenomenon could actually be relevant to model learning.  So, to me there does not seem to be easy fixes for this issue.

**2.** It is also unclear if the experiments demonstrate practically relevant improvement: the proposed method uses label information, but is the baseline a conditional model or unconditional one?  It would only be fair to compare to a conditional baseline but there is no mention of this issue in the text.

---

> ### Author Response · Authors · 2024-05-26
>
> ##    Weaknesses
> **Q1: I have doubts about the soundness of the motivation: generally speaking, the probability of two randomly sampled continuous paths crossing each other should quickly vanish as the dimensionality of the data space increases. This seems analogous to the fact that two random walks on $\mathbb{Z}^d$ will meet each other with probability 1 if $d\le 2$, and 0 if $d>2$. Therefore, it's quite unclear if the path crossing phenomenon should ever happen on higher-dimensional datasets such as image datasets...**
>
> A: We further dicuss the xflow in real life model in term of existence of crossing and redirection with different inference step. Please refer to **Comment on Common Question Q1** for details.
>
> **Q2: It is also unclear if the experiments demonstrate practically relevant improvement: the proposed method uses label information, but is the baseline a conditional model or unconditional one? It would only be fair to compare to a conditional baseline but there is no mention of this issue in the text.**
>
> A: All the models in this paper are uncondition to class label. For our model, we only use the prediction noise as condition to avoid crossing. Therefore, the comparisons are fair.
>
> ##    Requested Changes
> **Q1: Regarding (1) above, it would be helpful to demonstrate if exact path crossing happens in higher-dimensional problems through theoretical and numerical examples. I suggest having $d\ge 20$ dimensions with a similar scale of variance, in light of previous findings on the intrinsic dimensionality of image data (Pope et al, 2021).**
>
> A: We further dicuss the xflow in real life model in term of existence of crossing and redirection with different inference step. We have add Sec. D to the Appendix to futher discuss it.
>
> **Q2: Clarification on (2) above would be welcome.**
>
> A: We have added more details in Implementation Details in Sec. 4.2 to clarfy that all models in our paper are training without label as condition.

---

### Review · Reviewer_LiuX · 2024-05-19

**Summary Of Contributions:**

This paper tackles a common issue in diffusion model generation, that the generated trajectory may deviate from the original flow and follow an incorrect path. This paper attributes this issue to a phenomenon termed "xFlow," which is caused by the ambiguity and uncertainty in the target during the training process. To address this, the paper proposes increasing the dimension of the input with predicted noise in the early stage. The experimental results show that this method improves the consistency in the inference process, especially when the number of the sampling steps is small.

**Audience:**

Yes

**Broader Impact Concerns:**

This paper has no ethical implication concern.

**Claims And Evidence:**

No

**Requested Changes:**

1. Deeper analysis in Section 3.2.
- Whether DDPM (in design) really has such a issue？Existing claim appears speculative without robust evidence.
- Stronger connection with the method design.

If this issue is addressed, this paper would be strong.

2. Improve the clarity of the method design as stated in the weaknesses part. Address this would strengthen this work.

3. Extend the experiments to larger datasets to further demonstrate the effectiveness of the method. Address this would strengthen this work.

**Strengths And Weaknesses:**

# Strengths:

1. The discovered xFlow phenomenon is very interesting. This paper provides an optimization perspective on how the phenomenon is caused and proposes a reasonable method to solve it.
2. The experimental results show a significant improvement over the baseline method, especially with fewer inference steps.

# Weaknesses:

1. The discussion on how xFlow is caused in Section 3.2 is not very thorough. This section tries to attribute the causes of xFlow to the training process, that two training flows may overlap at certain time steps, but they have different targets. However, this doesn’t mean the targets are *incorrect*. The targets may be different, but the probability of each target is different. The diffusion model can learn that. Besides, the deduction from equations (8) and (9) to (10) is unclear, as $(x-z)^2 + (y-z)^2$ do not equal $((𝑥+𝑦)/2-z)^2$. A more clear and detailed analysis is needed.
2. The description of the method in Section 3.3 needs improvement in clarity.
- In “Selection of Condition”, the paper suggests that using the initial noise x_T for dimension ascending is more practical. However, in the “training stage", the paper says directly using the initial noise would cause trivial solutions, and replace it to the predicted noise. The paper should elaborate on why initial noise leads to trivial solutions and how using predicted noise could mitigates this issue.
- The paper proposes a bootstrap strategy to add robustness to the learning process and avoids misleading the predicted noise in the early stage. However, the explanation of how bootstrapping specifically contributes to robustness is not very clear.
- In the inference stage, to alleviate the computational costs, this paper uses the estimated noise in the previous step instead of the current step as the condition. The paper could provide analysis on the performance gap between using current step and previous step, and illustrate the trade-off between computational savings and the potential drop in generation quality.
3. To further demonstrate the effectiveness of the method, it is recommended to expand the experiments to larger datasets such as LSUN and CelebA. Additionally, exploring how to integrate this method to improve the prompt-following ability in text-to-image generation would be beneficial.

---

> ### Author Response · Authors · 2024-05-26
>
> ##    Weakness
> **Q1：The discussion on how xFlow is caused in Sec. 3.2 is not very thorough...**
>
> A：Thank you for your advice. We acknowledge that the description in Sec. 3.2 might lead to confusion, thus we have revised this section for clarity. For more details, please refer to Sec. 3.2.
>
> **Q2(1): In “Selection of Condition” ... The paper should elaborate on why initial noise leads to trivial solutions and how using predicted noise could mitigates this issue.**
>
> A:In the training stage of diffusion model, the loss function is:
> \begin{gather}
>     L_{simple} = E_{t,x_0,\epsilon}[\| \epsilon - \epsilon_\theta(x_t,t)\|^2],
> \end{gather}
> If we use initial noise as condition, then the loss function will be:
> \begin{gather}
>     L_{simple} = E_{t,x_0,\epsilon}[\| \epsilon - \epsilon_\theta(x_t,\epsilon,t)\|^2],
> \end{gather}
> Therefore, during training, since we give the target $\epsilon$ as input, the model $\epsilon_\theta(x_t,\epsilon,t)$ will directly ouput $\epsilon$ without learning anything. We have run an experiment with above setting. The $L_{simple}$ will quickly descend to near 0, however, the generated images are almost noise without any useful semantic information, which denotes that using initial noise as condition will leads to trival solution.
>
> By using predicted noise as condition, the loss function is:
> \begin{gather}
> L_{simple} = E_{t,x_0,\epsilon}[\| \epsilon - \epsilon_\theta(x_t,\hat{\epsilon},t)\|^2], \\\\
> \hat{\epsilon} = \epsilon_\theta(x_t,\mathbf{0},t).
> \end{gather}
> Therefore, we prevent to use the target $\epsilon$ as input of the diffusion model and hence avoid trival solution.
>
> **Q2(2): The paper proposes a bootstrap strategy to add robustness to the learning process and ...**
>
> A: In training, we replace $\epsilon$ with $\hat{\epsilon}$. Therefore, we expect that the gap between $\epsilon$ and $\hat{\epsilon}$ is small. However, in the early stage of training, $\|\epsilon-\hat{\epsilon}\|^2$ is large, so we use bootstrap to prevent continuously using bad condition. The ablation study in Table 4 also demonstrate that without bootstrap strategy, model will generate images with poor quality. Such a strategy is also recognized as *annealing*, which is commonly used in training neural networks.
>
> **Q2(3): In the inference stage, to alleviate the computational costs, ...**
>
> Table 1 shows how performance varies between using the current and previous steps in the inference process. When the inference step is large and step size small (e.g., 1000), the predicted noise and resulting quality are similar for both steps. However, with a small inference step and large step size (e.g., 5), significant differences in predicted noise lead to poor performance when using the previous step.
>
> In terms of trade-off, if the model use predicted noise in current step, the inference will be
>
> \begin{gather}
> \hat{\epsilon} = \epsilon_\theta(x_t,\mathbf{0},t). \\\\
> \hat{\epsilon_t}= \epsilon_\theta(x_t,\hat{\epsilon},t)
> \end{gather}
>
> Comparing with model using previous step as Eq.22, the inference cost will be double.
>
> **Q3: To further demonstrate the effectiveness of the method, it is recommended to expand the experiments to larger datasets such as LSUN and CelebA...**
>
> A: Currently, we do not have access to the resources to train diffusion models on larger datasets and leave this as future work. Nevertheless, our visualizations of Stable Diffusion's inference flow in Fig.9 show frequent xFlow occurrences, which leads to sub-optimal or OOD synthesis. We hope these findings will inspire further research.
>
> ##    Requested Changes
>
> **Q1(1): Whether DDPM (in design) really has such a issue？...**
>
> A: We have added Sec. D to the Appendix to further shows the xFlow in real life model and dataset.
>
> **Q1(2): Stronger connection with the method design.**
>
> A: The connection between the analysis and our model design is discussed in the second and the third paragraph in Sec. 3.3. To emphasize the connection, we have revised that part.
>
> **Q2: Improve the clarity of the method design as stated in the weaknesses part...**
> * For Q1, we have revised Sec. 3.2 to further discuss why crossing will lead to *incorrect target*.
> * For Q2(1) , we have added further discussion about trivial solutions with initial noise as condition in Sec. C.
> * For Q2(2), we have revised the Training Stage in Sec. 3.3 to clarify why bootstrapping  contributes to robustness.
> * For Q2(3), the analysis on the performance gap between using current step and previous step is dicussed in Comparison of Sampling Quality in Sec. 4.2.
> * For Q3, we have added visualization of Stable Diffusion in Sec. D.
>
>
> **Q3: Extend the experiments to larger datasets to further demonstrate the effectiveness of the method...**
>
> We understand that the extension could strengthen this work. However, currently, we do not have access to the resources for training diffusion models on larger dataset. We leave these experiments as future work.

---

> > ### Comment · Reviewer_LiuX · 2024-06-19
> > **Response by Reviewer LiuX**
> >
> > Thanks for the detailed response from the authors. The explanations provided have alleviated my concerns about the method design. However, regarding the existence of the xFlow phenomenon in real-life models, the response still lacks robust evidence. Although the authors have added a discussion in Section D of the appendix, Proposition D.2 remains unproven. Merely proving that n2 follows a standard normal distribution does not lead to the conclusion that the existence of n2 = n1 +  [t/(t-1)] (x1-x2). Furthermore, the discussion on Stable Diffusion does not directly answer whether the xFlow phenomenon truly exists. Figure 9 only demonstrates that under different inference timesteps, Stable Diffusion exhibits different directions, which can be attributed to increased prediction error with reduced steps rather than the xFlow phenomenon. This unresolved issue represents a main weakness of the paper.

---

> ### Author Response · Authors · 2024-06-20
>
> Thank you for the comments. We will elaborate on the existence of the xFlow phenomenon in real-life models.
>
> For Proposition D.2, $n_2$ is exactly calculated by $n_1 + [t/(t-1)] (x_1-x_2)$, i.e., $n_2 \triangleq n_1 + [t/(t-1)] (x_1-x_2)$. We then aim to demonstrate that $n_2$ can be sampled from the distribution of noise, specifically following the standard normal distribution, through empirical testing.
>
> Regarding the discussion on Stable Diffusion, it is reasonable to attribute the redirection to xFlow rather than merely a reduced step. The ambiguity caused by xFlow contributes to prediction error, as analyzed in Section 3.2, where we show that xFlow affects the training process. Therefore, the existence of xFlow does not conflict with prediction error. In fact, prediction error is caused by both xFlow and reduced steps. Our model shows semantic consistency even with reduced steps compared to the baseline model. A similar result is observed in Rectified Flow[1]: For 1-Rectified Flow, where crossings still exist, reduced steps lead to redirection. However, for 2-Rectified Flow without crossings, the model does not suffer from redirection even with reduced steps. These experiments imply that the presence of xFlow leads to redirection, while suppressing xFlow reduces redirections. Therefore, if a model suffers from redirections, it must have xFlow. The reduced step is only a variable to observe this phenomenon, rather than a direct cause. Thus, the redirection in Stable Diffusion demonstrates that xFlow also exists in real-life models.
>
> In conclusion, xFlow exists in real-life models like Stable Diffusion. We hope our explanation alleviates your concerns.
>
> [1] Liu X, Gong C, Liu Q. Flow straight and fast: Learning to generate and transfer data with rectified flow. ICLR, 2023.

---

### Author Response · Authors · 2024-05-26
**Comment on Common Question**

**Q1:  Does the xFlow phenomenon appears in real life models or high-dimensional space?**

A: Yes, the xFlow phenomenon appears in real life models and high-dimensional space.

**Existence of crossing in high-dimensional space**. To demonstrate the existence of crossing during training with high-dimensional data, we formulate the problem with the following propositions:

*Proposition 1 (existence of crossing point):* Given images or latents $x_1$, $x_2$, randomly sample noise $n_1\sim \mathcal{N}(\mathbf{0}, \mathbf{I})$, there exists timestep $t\in[0,1]$ and noise $n_2\sim \mathcal{N}(\mathbf{0}, \mathbf{I})$ s.t. $x_1^{t}=x_2^{t}$, where $x_1^{t}=tx_1 + (1-t)n_1$ and $x_2^{t}=tx_2 + (1-t)n_2$.

Since during the training stage of diffusion model, the timestep is sampled from a fixed set $\mathcal{S}$, like $[\frac{1}{n},\frac{2}{n},...,\frac{n-1}{n}]$, the above proposition can be equivalent:

*Proposition 2 (existence of crossing point with discrete timesteps)*: Given images or latents $x_1$,$x_2$, randomly sample noise $n_1\sim \mathcal{N}(\mathbf{0}, \mathbf{I})$, there exists timestep $t\in\mathcal{S}$ s.t. $n_2=n_1+\frac{t}{1-t}(x_1-x_2)$ and $n_2\in\mathcal{N}(\mathbf{0}, \mathbf{I})$.

Note that $x_1,x_2$ are given constant vectors, therefore, the term $\frac{t}{1-t}(x_1-x_2)$ does not change the covariance of noise $n_1$. Thus, the verification challenge reduces to demonstrating that all the elements of $n_2$ follow $\mathcal{N}(0,1)$.  However, formally proving Proposition 2 is still challenging, as the distribution of $n_2$ is highly related to the distribution of dataset. Due to these complexities, we opt for empirical verification of the crossing phenomenon within specific datasets.


We sampled 50k image pairs $(x_1, x_2)$ from the CIFAR-10 dataset, along with noise $n_1 \sim \mathcal{N}(\mathbf{0}, \mathbf{I})$. We traversed timesteps $t$ in the set $\mathcal{S} =  [ \frac{999}{1000}, \frac{998}{1000}, \ldots, \frac{100}{1000} ] $ to compute $n_2$.
For the Shapiro-Wilk test, we set the alpha level at $\alpha = 0.05$. We assume $n_2$ follows a standard normal distribution if it satisfies the following conditions:

* The p-value exceeds 0.05.
* The mean falls within $[0 - E_m, 0 + E_m]$, where $E_m = 0.015$.
* The standard deviation lies within $[1 - E_s, 1 + E_s]$, where $E_s = 0.01$.


The parameters $E_m$ and $E_s$ are set based on the average absolute error in mean and standard deviation from similarly scaled randomly sampled data.


The experiment suggests that the existence ratio of crossing in Cifar-10 is **19.64\%**, which means, randomly sample an image pair in Cifar-10, with probability 19.64\%, there exists noise pairs that make the training flow cross. This implies that despite the complexities inherent in high-dimensional spaces, crossing phenomena still manifest during the training stages of models.

**Redirection in real life models**. We investigated the inference flow of the text-to-image model, Stable Diffusion, through various stages. By  visualizing the intermediate latent representations with t-SNE at each step, we observed substantial redirection in the inference process. Figure 9 in the revised draft illustrates this phenomenon, highlighting the significant influence of xFlow on the model's behavior during different inference steps.

---

### Decision · Action_Editor_ksj1 · 2024-07-04

**Recommendation:** Reject

**Comment:**

Reviewers highlighted that the paper lacks convincing real-world evidence demonstrating the xFlow phenomenon and raised concerns about its low probability of occurrence in high-dimensional spaces. The authors' response, which included a proposition about noise variables, did not adequately address these concerns and lacked formal proof. Consequently, the claims made in the submission are not well supported by accurate, convincing, and clear evidence.

**Audience:**

Some individuals in TMLR's audience might find the findings of this paper interesting. Despite the concerns about the lack of sufficient evidence supporting the xFlow phenomenon, the concept itself looks quite interesting.

**Claims And Evidence:**

Based on the reviewers' comments, the claims made in the submission are not supported by accurate, convincing, and clear evidence. The xFlow phenomenon lacks sufficient real-world examples, and its occurrence in high-dimensional spaces is highly questionable. The authors' response, including a proposition about noise variables, does not address the core concern that the phenomenon may occur with extremely low probability. Additionally, the absence of formal proof and reliance on empirical verification further weaken the credibility of the claims.

**Resubmission Of Major Revision:**

The authors may consider submitting a major revision at a later time.